DOI: 10.1038/s41467–017–01466-8    **OPEN**

# Real-space and real-time dynamics of CRISPR-Cas9 visualized by high-speed atomic force microscopy

Mikihiro Shibata[1,2], Hiroshi Nishimasu[3,4], Noriyuki Kodera[2,4], Seiichi Hirano[3], Toshio Ando[2,5], Takayuki Uchihashi[2,5,6,7] & Osamu Nureki [3]

The CRISPR-associated endonuclease Cas9 binds to a guide RNA and cleaves double-stranded DNA with a sequence complementary to the RNA guide. The Cas9–RNA system has been harnessed for numerous applications, such as genome editing. Here we use high-speed atomic force microscopy (HS-AFM) to visualize the real-space and real-time dynamics of CRISPR-Cas9 in action. HS-AFM movies indicate that, whereas apo-Cas9 adopts unexpected flexible conformations, Cas9–RNA forms a stable bilobed structure and interrogates target sites on the DNA by three-dimensional diffusion. These movies also provide real-time visualization of the Cas9-mediated DNA cleavage process. Notably, the Cas9 HNH nuclease domain fluctuates upon DNA binding, and subsequently adopts an active conformation, where the HNH active site is docked at the cleavage site in the target DNA. Collectively, our HS-AFM data extend our understanding of the action mechanism of CRISPR-Cas9.

[1] High-speed AFM for Biological Application Unit, Institute for Frontier Science Initiative, Kanazawa University, Kakuma, Kanazawa 920-1192, Japan. [2] Bio-AFM Frontier Research Center, Kanazawa University, Kakuma, Kanazawa 920-1192, Japan. [3] Department of Biological Sciences, Graduate School of Science, The University of Tokyo, 2-11-16 Yayoi, Bunkyo-ku, Tokyo 113-0032, Japan. [4] JST, PRESTO, 4-1-8 Honcho, Kawaguchi, Saitama 332-0012, Japan. [5] CREST/JST, 4-1-8 Honcho, Kawaguchi, Saitama 332-0012, Japan. [6] Department of Physics, Kanazawa University, Kakuma, Kanazawa 920-1192, Japan. [7] Present address: Department of Physics, Nagoya University, Furo-cho, Chikusa-ku, Nagoya 464-8602, Japan. Mikihiro Shibata and Hiroshi Nishimasu contributed equally to this work. Correspondence and requests for materials should be addressed to M.S. (email: msshibata@staff.kanazawa-u.ac.jp) or to H.N. (email: nisimasu@bs.s.u-tokyo.ac.jp) or to O.N. (email: nureki@bs.s.u-tokyo.ac.jp)

In the microbial CRISPR-Cas (clustered regularly interspaced short palindromic repeats-CRISPR-associated proteins) adaptive immune system[1–3], the RNA-guided DNA endonuclease Cas9 associates with a CRISPR RNA (crRNA) and a *trans*-activating crRNA (tracrRNA), and cleaves the double-stranded (ds) DNA complementary to the crRNA guide[4–7] (Supplementary Fig. 1a). Cas9 can function with a single-guide RNA (sgRNA), in which the crRNA and the tracrRNA are fused with an artificial linker[7]. The two-component system, consisting of Cas9 from *Streptococcus pyogenes* and the sgRNA, has been harnessed for genome-engineering technologies in a variety of cell types and organisms[8–10]. Cas9–RNA first recognizes a short nucleotide sequence (NGG for *S. pyogenes* Cas9; N represents any nucleotide) next to a target sequence in the dsDNA, called a protospacer adjacent motif (PAM). Cas9–RNA then initiates DNA unwinding at the PAM-proximal region, which is followed by the directional formation of an R-loop, consisting of the RNA–DNA hybrid and the displaced non-target DNA strand[11]. After the R-loop formation, the HNH and RuvC nuclease domains of Cas9 cleave the target DNA strand (complementary to the RNA guide) and the non-target DNA strand, respectively[6,7].

Previous structural studies showed that apo-Cas9 adopts a closed auto-inhibited conformation[12], whereas the Cas9–RNA binary complex adopts a bilobed architecture comprising an α-helical recognition (REC) lobe and a nuclease (NUC) lobe[13] (Fig. 1a, b). A comparison between these structures suggests that Cas9 undergoes a closed-to-open structural rearrangement upon binding the guide RNA. Moreover, the crystal structures of Cas9–RNA bound to the target DNA provided mechanistic

insights into RNA-guided DNA targeting by Cas9[14,15] (Fig. 1b). Furthermore, the Cas9 R-loop complex structure revealed drastic conformational changes in the linker regions between the RuvC and HNH nuclease domains, thereby translocating the HNH domain closer to the target DNA strand[16] (Fig. 1b, Supplementary Fig. 1b). Consistently, bulk and single-molecule Förster resonance energy transfer (FRET) studies indicated that the HNH domain undergoes a structural transition during DNA cleavage and adopts three major conformations: RNA-bound (R), intermediate (I) and active docked (D) states[17,18]. The R and I conformations predominantly correspond to the crystal structures of Cas9–RNA[13] and Cas9–RNA–DNA[14,15], respectively. A structure of the D conformation was predicted by modeling[17,18], but it has not been determined, although the D conformation approximates the crystal structure of the Cas9 R-loop complex[16]. These structural and imaging studies provided mechanistic insights into the Cas9-mediated DNA recognition and cleavage, but its action mechanism has not been fully clarified. It is unknown how apo-Cas9 in the closed conformation assembles with the guide RNA to form an effector complex. In addition, Cas9 in the catalytically-active D conformation has not been visualized.

High-speed atomic force microscopy (HS-AFM) is a powerful technique that enables real-space and real-time observations of macromolecules, which are not feasible by other techniques[19]. HS-AFM imaging has elucidated the dynamics of various proteins with unprecedented details; for example, the photo-induced conformational change of bacteriorhodopsin[20], myosin V walking on an actin filament[21], cellulose degradation by a cellulase

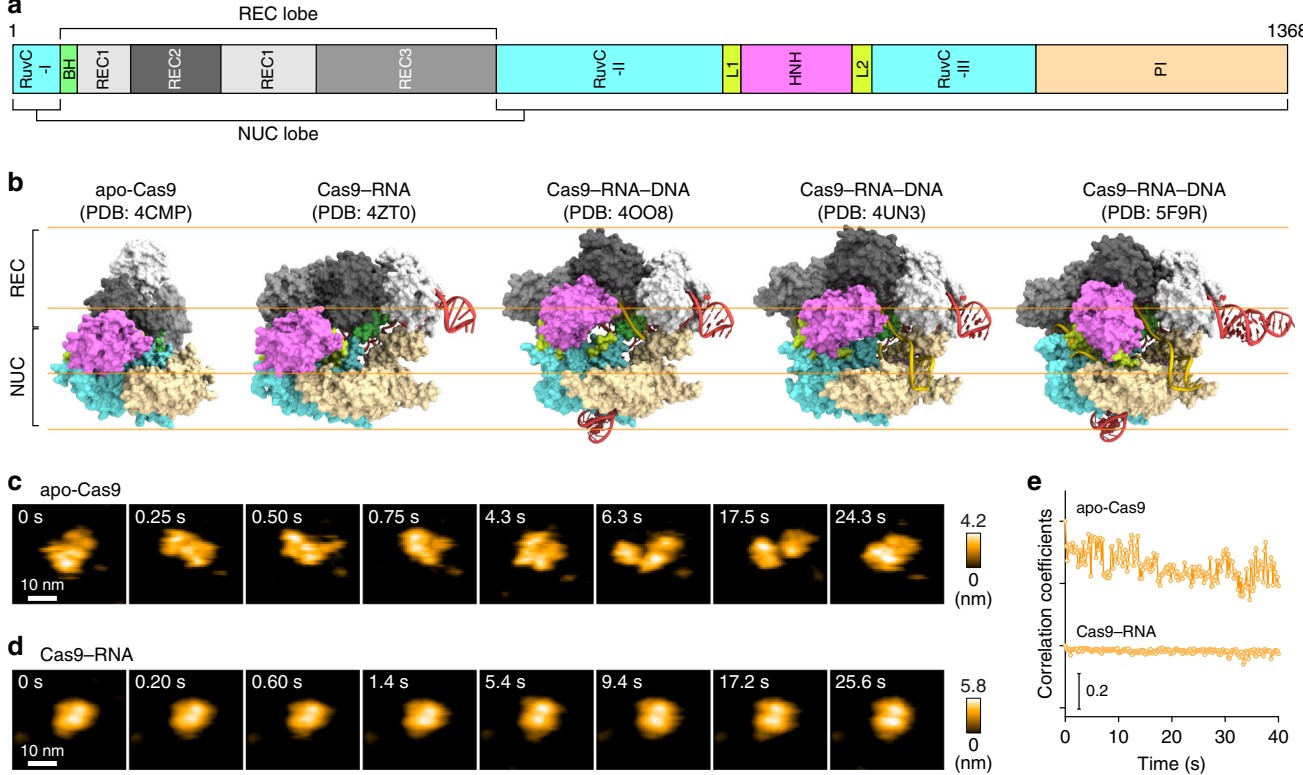

**Fig. 1** HS-AFM observations of apo-Cas9 and Cas9–RNA. **a** Domain structure of *S. pyogenes* Cas9. BH, Bridge helix. **b** Crystal structures of apo-Cas9 (PDB: 4CMP)[12], Cas9–RNA (PDB: 4ZT0)[13], Cas9–RNA bound to its single-stranded DNA target (PDB: 4OO8)[15], Cas9–RNA bound to a partial DNA duplex (PDB: 4UN3)[14] and Cas9–RNA bound to its dsDNA target (a Cas9 R-loop complex) (PDB: 5F9R)[16]. The guide RNA and the target DNA are colored red and yellow, respectively. The PAM is colored purple. The 98-nt guide RNA (PDB: 4OO8) was used for HS-AFM observations. **c, d** Sequential HS-AFM images of apo-Cas9 (**c**) and Cas9–RNA (**d**) on the AP-mica surface. The color (from black to white) corresponds to the height. The scale bars are 10 nm. **e** Time courses of correlation coefficients between the sequential HS-AFM images of apo-Cas9 and Cas9–RNA

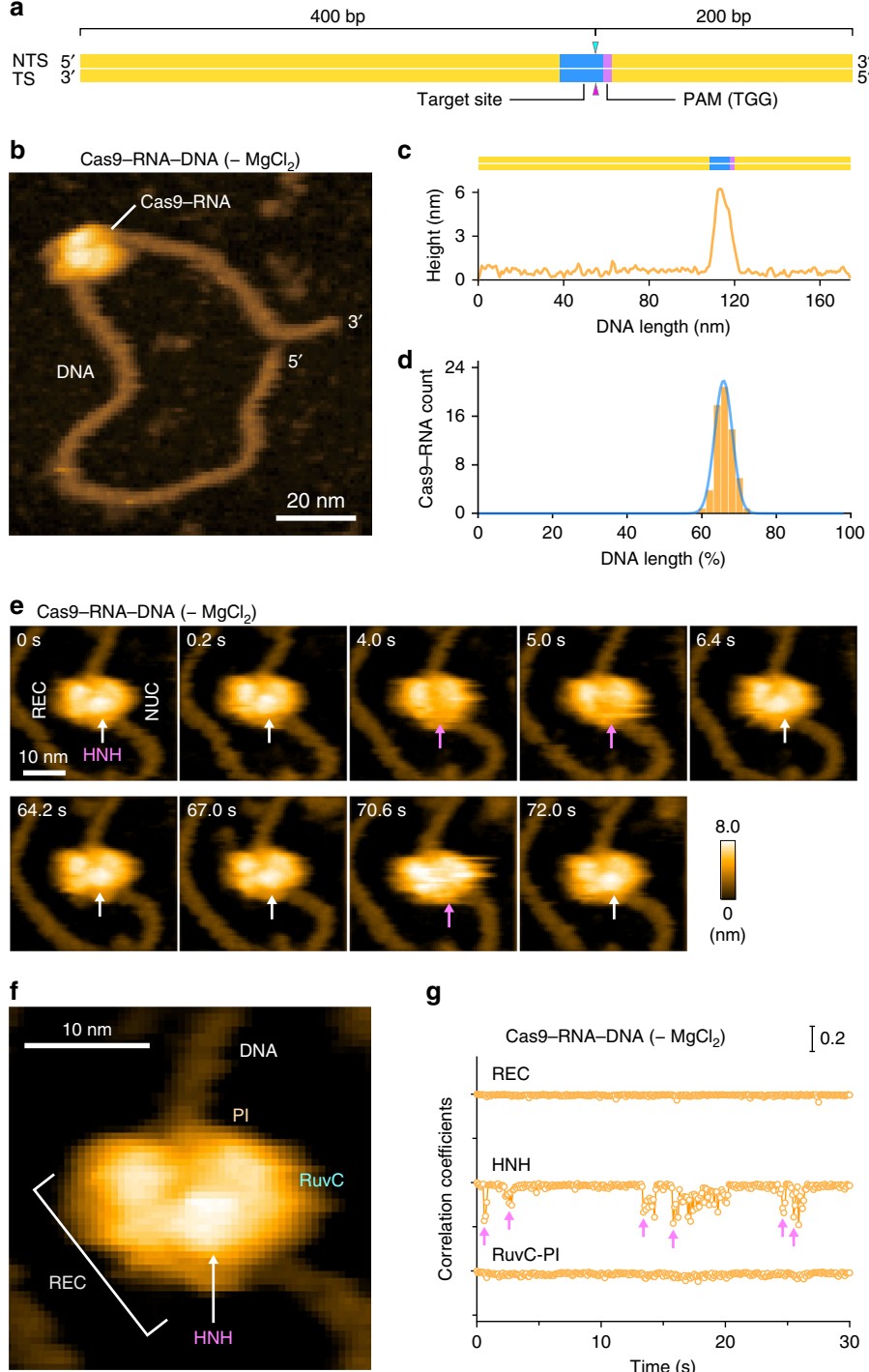

**Fig. 2** HS-AFM observations of Cas9–RNA–DNA. **a** Schematic of the dsDNA substrate. The target site and the PAM are colored blue and purple, respectively. The sites cleaved by the RuvC and HNH domains are indicated by the cyan and magenta triangles, respectively. TS, target strand; NTS, non-target strand. **b** HS-AFM image of Cas9–RNA–DNA in the absence of MgCl₂. The scale bar is 20 nm. **c** Cross-sectional profile along the DNA in a representative HS-AFM image of Cas9–RNA–DNA. **d** Distribution of the height peaks in the HS-AFM images of Cas9–RNA–DNA ($n = 65$). The peak distribution fits a Gaussian curve, with the peak corresponding to the target site. **e** Sequential HS-AFM images of Cas9–RNA–DNA in the absence of MgCl₂. The HNH domain is indicated by white arrows, whereas its disappearance (fluctuation) is indicated by magenta arrows. The scale bar is 10 nm. **f** Close-up view of a representative HS-AFM image of Cas9–RNA–DNA. The scale bar is 10 nm. **g** Time courses of correlation coefficients for the individual domains between the sequential HS-AFM images of Cas9–RNA–DNA in the absence of MgCl₂. The HNH domain fluctuations are indicated by magenta arrows

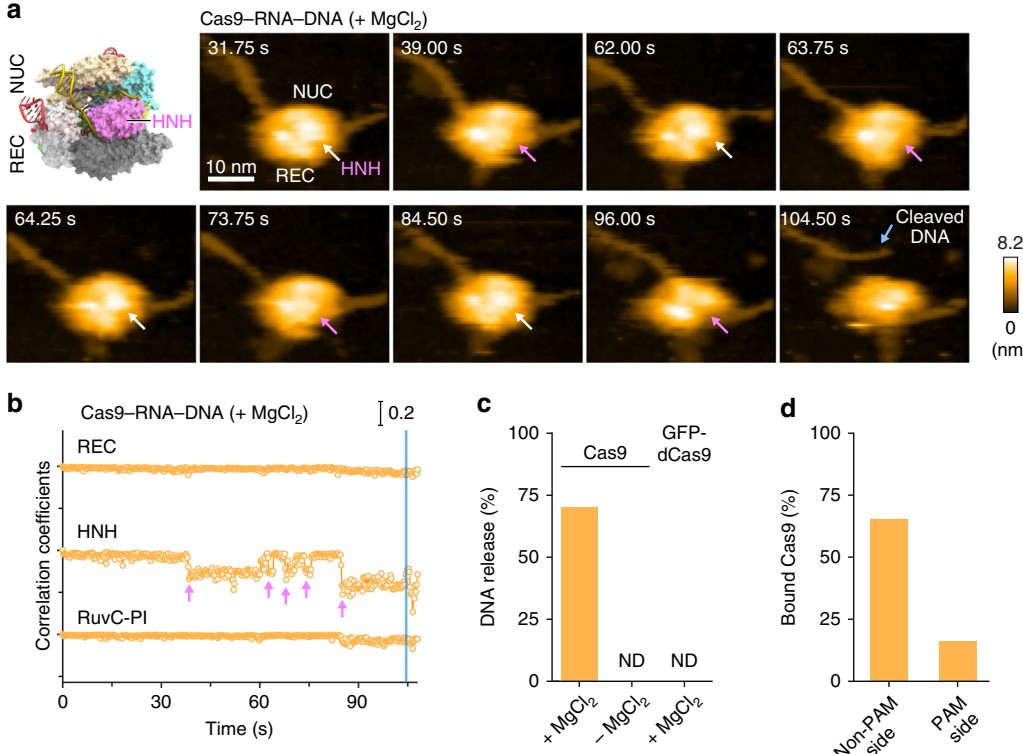

**Fig. 3** HS-AFM observations of DNA cleavage by Cas9–RNA. **a** Sequential HS-AFM images of Cas9–RNA–DNA in the presence of MgCl$_2$. The HNH domains in the inactive (high-height) and active (low-height) states are indicated by white and magenta arrows, respectively. The scale bar is 10 nm. **b** Time courses of correlation coefficients for the individual domains between the sequential HS-AFM images of Cas9–RNA–DNA in the presence of MgCl$_2$. The HNH domain fluctuations are indicated by magenta arrows. The release of the cleavage product is indicated by a blue line. **c** Rates of the cleavage product release from Cas9–RNA in the presence ($n = 361$) and absence ($n = 36$) of MgCl$_2$, and from GFP-dCas9–RNA ($n = 37$). ND, not detected. **d** Binding position of Cas9–RNA after the product release ($n = 181$)

enzyme[22], rotary catalysis of F$_1$-ATPase[23], lipid membrane remodeling by ESCRT-III polymerization[24] and lipid membrane stabilization by annexin V[25].

In this study, we employed HS-AFM to visualize the real-space and real-time dynamics of CRISPR-Cas9 in action, at nanometer resolution. HS-AFM movies revealed that apo-Cas9 adopts flexible conformations, whereas Cas9–RNA forms a stable bilobed architecture. Furthermore, the HS-AFM movies directly visualized the Cas9-mediated DNA cleavage reaction accompanied by the drastic structural transition of the HNH nuclease domain. Overall, the HS-AFM movies provided distinct scenes of Cas9 in action, comprising the complex assembly, target search, target cleavage and product release, thus substantially improving our mechanistic understanding of CRISPR-Cas9.

## Results

**RNA-induced structural stabilization in Cas9.** We first observed apo-Cas9 and pre-assembled Cas9–RNA on a mica surface treated with 3-aminopropyl-triethoxysilane (AP-mica). Unexpectedly, the HS-AFM movies revealed that apo-Cas9 adopts flexible modular conformations, unlike the stable closed conformation observed in the crystal structure[12] (Fig. 1b, c, Supplementary Movie 1). In contrast, the HS-AFM movies of Cas9–RNA showed a stable bilobed architecture, consistent with the crystal structure[13] (Fig. 1b, d, Supplementary Movie 2). The correlation coefficients for the sequential HS-AFM images highlighted the substantial differences in the conformational flexibilities between apo-Cas9 and Cas9–RNA (Fig. 1e, Supplementary Fig. 2a, b). A structural comparison between apo-Cas9[12] and Cas9–RNA[13] indicated that the three domains (REC1–3) in

the REC lobe adopt distinct arrangements, whereas the RuvC domain interacts similarly with the HNH and PAM-interacting (PI) domains to form the NUC lobe structure (Fig. 1b). This supports the notion that the three REC domains of apo-Cas9 adopt flexible conformations in solution, although apo-Cas9 adopted a closed conformation in the crystal structure, probably due to crystal packing interactions. Together, our HS-AFM data reveal the unexpected conformational flexibility of apo-Cas9, and highlight the guide-RNA-mediated stabilization of the REC lobe conformation and induction of structural rearrangements in the Cas9 protein.

**PAM-dependent DNA targeting by Cas9–RNA.** We next sought to visualize the binding of Cas9–RNA to the target DNA. To avoid Mg$^{2+}$-dependent DNA cleavage by Cas9[7], we incubated the pre-assembled Cas9–RNA and a 600-bp dsDNA containing a 20-nt target site with the TGG PAM 400-bp downstream from its 5′ end, in the absence of Mg$^{2+}$ (Fig. 2a). We then adsorbed the Cas9–RNA–DNA complex on the AP-mica surface, and performed HS-AFM observations. The HS-AFM movies revealed that Cas9–RNA specifically binds to the expected target site in the DNA (Fig. 2b, c, Supplementary Movie 3). An analysis of the HS-AFM images confirmed the specific binding of Cas9–RNA to the target site in all of the observed DNA molecules (Fig. 2d, Supplementary Fig. 3a). In contrast, Cas9–RNA did not bind to the target DNA containing TTT, rather than TGG, as the PAM (Supplementary Fig. 3b), consistent with the observation that Cas9 requires the NGG sequence as the PAM for DNA recognition[7,11]. These results demonstrate that our HS-AFM movies

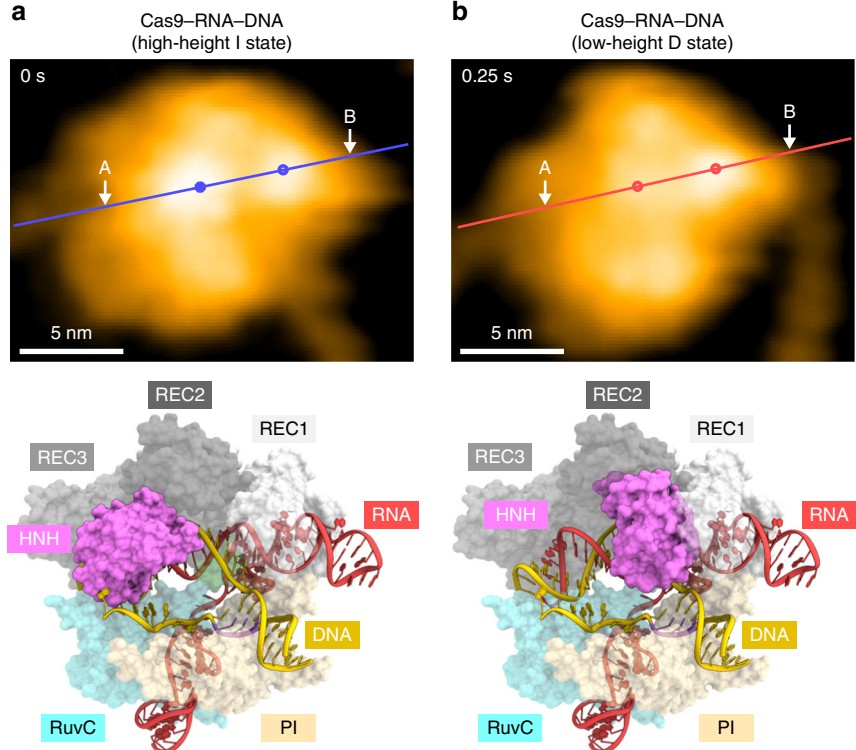

**Fig. 4** Structural rearrangement of the HNH domain. **a**, **b** HS-AFM images of the HNH domain in the high-height (**a**) and low-height (**b**) states. The mean center positions of the HNH and REC1 domains are indicated by dots. Red and blue lines indicate the cross-sectional position used for the height distribution analysis in Supplementary Fig. 5d. For comparison, the Cas9–RNA–DNA models in the I and D states are shown below the respective images. The structural models consist of Cas9–RNA (PDB: 4OO8) and DNA (PDB: 5F9R), and the D state model was built as described previously[17]. The scale bars are 5 nm

faithfully recapitulate the PAM-dependent target recognition by Cas9–RNA.

In the HS-AFM movies of Cas9–RNA–DNA, we observed a prominent protrusion between the two lobes, which is not discernible in the Cas9–RNA movies (Fig. 2e, Supplementary Movie 3). A comparison with the crystal structures[13–16] indicated that this protrusion corresponds to the HNH nuclease domain (Figs. 1b and 2f). The domain assignment is further supported by the HS-AFM images of N-terminal GFP-fused dCas9(D10A/H840A)–RNA bound to the DNA (Supplementary Fig. 3c, d, Supplementary Movie 4). We observed that Cas9–RNA binding induces ~ 30° local bending in the target DNA, consistent with the crystal structures of Cas9–RNA–DNA[14,16]. Notably, the protrusion frequently disappeared for a short time during the HS-AFM imaging (Fig. 2e, Supplementary Movie 3). A time course of the correlation coefficients calculated for a limited area on the three regions (REC, HNH and RuvC-PI) showed that the HNH domain fluctuates in the Cas9–RNA–DNA complex, unlike the other domains (Fig. 2g, Supplementary Fig. 3e). Thus, these HS-AFM data provide direct visualizations of the conformational dynamics of the HNH domain upon DNA binding, as suggested by previous structural studies[15,16] and FRET experiments[17,18].

**Target DNA cleavage by Cas9–RNA.** We next sought to observe the target DNA cleavage by Cas9–RNA. To this end, we mixed pre-assembled Cas9–RNA with the target DNA in the absence of $Mg^{2+}$, adsorbed the complex on the AP-mica surface, and then initiated the cleavage reaction by the addition of $Mg^{2+}$. The HS-AFM movies revealed that the HNH domain also fluctuates in the presence of $Mg^{2+}$ (Fig. 3a, b, Supplementary Fig. 4,

Supplementary Movie 5). Notably, in the presence of $Mg^{2+}$, the HNH domain remained in a low-height state after several fluctuations, followed by the release of the DNA from the Cas9–RNA complex (Fig. 3a, b, Supplementary Fig. 4, Supplementary Movie 5). The DNA release was not observed in the absence of $Mg^{2+}$ (Fig. 3c). We observed the binding of nuclease-inactive dCas9–RNA to the target DNA, but the DNA was not released from the complex (Fig. 3c). These results confirmed that the released DNA represents a cleavage product, and indicated that, in the low-height state, the HNH active site is located near the scissile phosphate of the target strand to accomplish the DNA cleavage.

**Conformational dynamics of the HNH domain.** In the available Cas9–RNA–DNA structures, the HNH domain adopts catalytically inactive conformations and is not located near the scissile phosphate in the target DNA strand[14–16] (Supplementary Fig. 5a), suggesting that the HNH domain must undergo structural rearrangements to approach the cleavage site. Consistently, bulk and single-molecule FRET studies indicated that the HNH domain adopts three major conformations: R, I, and D states[17,18]. The R and I conformations are consistent with the crystal structures of the Cas9–RNA[13] and Cas9–RNA–DNA[14,15] complexes, respectively (Supplementary Fig. 5a). A structure in the D conformation has not been determined, but was predicted by modeling[17] (Supplementary Fig. 5b). In addition, structural and functional studies revealed that the L1 and L2 linker regions between the HNH and RuvC domains play a pivotal role in the conformational rearrangements of the HNH domain[15–17] (Supplementary Fig. 5c). Notably, the high- and low-height states observed in our HS-AFM images are in agreement with the I and

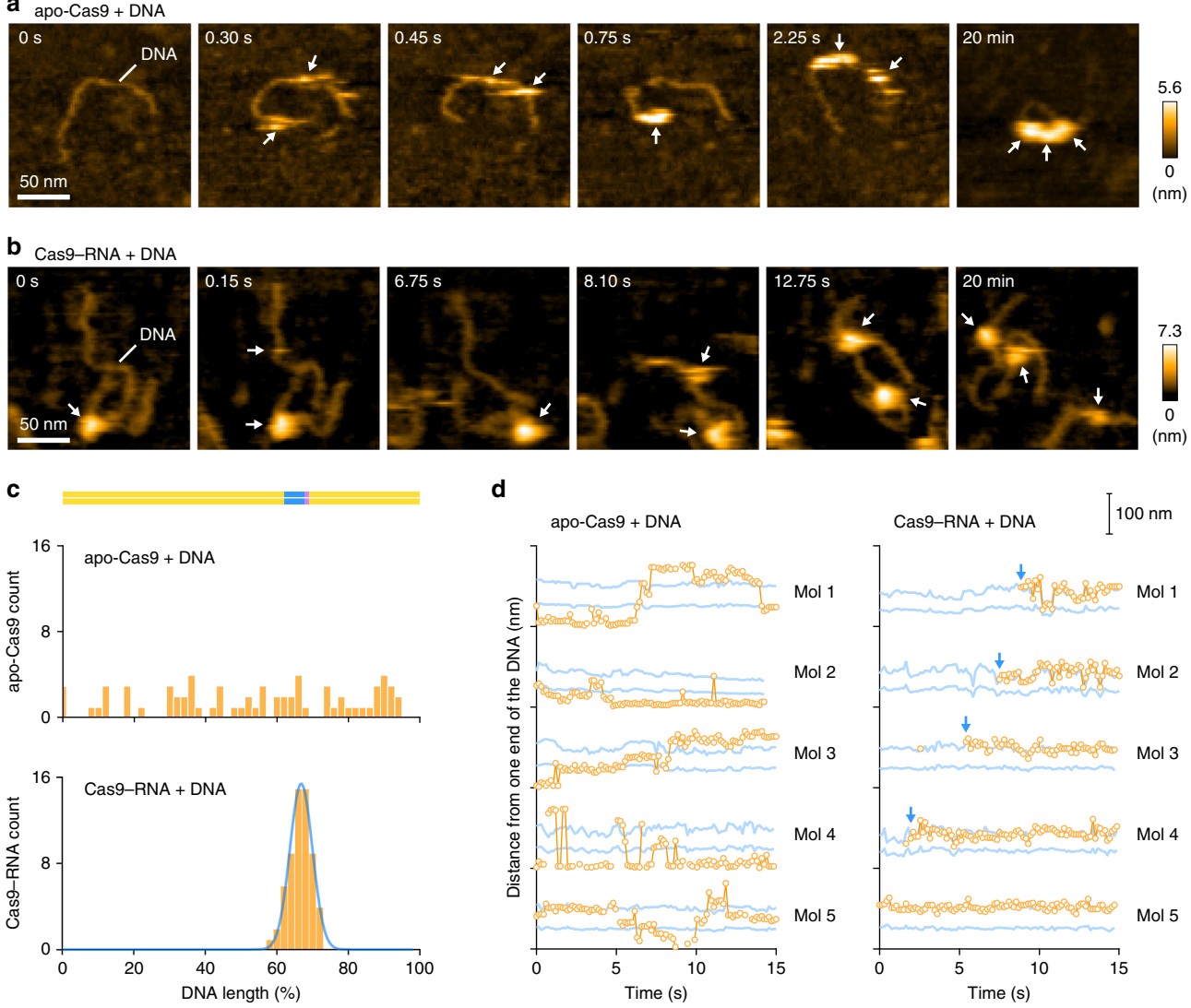

**Fig. 5** HS-AFM observations of target interrogation by Cas9–RNA. **a**, **b** Sequential HS-AFM images of the DNA after addition of apo-Cas9 (**a**) and Cas9–RNA (**b**) on the lipid bilayer. Apo-Cas9 and Cas9–RNA are indicated by white arrows. The scale bars are 50 nm. **c** Binding distributions of apo-Cas9 ($n = 69$) and Cas9–RNA ($n = 61$). The binding distribution of Cas9–RNA fits a Gaussian curve, with the peak corresponding to the target site. **d** Time courses of the binding positions of apo-Cas9 and Cas9–RNA. The distances from one end of the DNA were measured for five representative apo-Cas9 (left) and Cas9–RNA (right) molecules. Blue lines indicate the positions 200 and 400 bp from one end of the DNA (the potential target sites). Blue arrows indicate the binding of Cas9–RNA to the target site

D conformations, respectively (Fig. 4a, b). The height differences of the HNH domain in the two states ($0.8 \pm 0.2$ nm, $n = 14$) are likely to reflect the HNH displacement toward the target DNA for the cleavage reaction (Supplementary Fig. 5d). Thus, our HS-AFM movies directly visualized the catalytically active D state of Cas9, and revealed the conformational dynamics of the HNH domain during DNA cleavage.

**DNA release after cleavage**. Our HS-AFM movies revealed that most of the Cas9–RNA molecules remain bound to the PAM-distal region (the non-PAM side) of the cleaved DNA after the release of the PAM-containing region (the PAM side) (104.50 s; Fig. 3a, d, Supplementary Fig. 6a, b, Supplementary Movie 5). The dwell time of the low-height state before the DNA release ranged widely from 0.4 to 29.2 s, whereas previous biochemical experiments showed that Cas9–RNA remains tightly bound to the DNA even after cleavage[11,26]. This discrepancy suggests that the physical contacts with the AFM probe facilitate the dissociation of

Cas9–RNA from the DNA after cleavage. We observed some Cas9–RNA molecules that remained bound to the PAM side of the cleaved DNA after the release of the non-PAM side (Fig. 3d). Although this would require the unwinding of the RNA–DNA heteroduplex, it is unclear how the non-PAM side is released from the complex, due to the limited resolution of the HS-AFM imaging. The release of the non-PAM side was not observed in a previous DNA-curtain assay[11], and this discrepancy may also be derived from the effects of the contacts with the AFM probe. The HS-AFM movies showed that the released DNAs on the PAM side are apparently longer by ∼ 2.7 nm ($n = 14$), as compared with those before the release (Supplementary Fig. 6b, c). Given that the 8-bp PAM DNA duplex (0.34 nm/bp × 8 bp = 2.7 nm) is accommodated between the REC1 and PI domains in the crystal structure[14] (Supplementary Fig. 6d), this apparent extension of the PAM-side DNA is likely due to the release of the PAM-containing region, which is bound inside the Cas9–RNA molecule before the release.

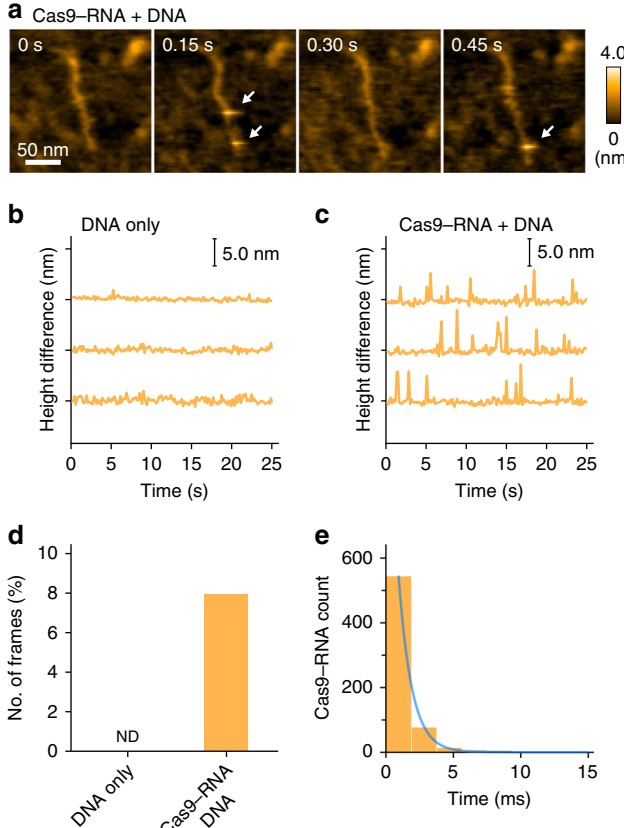

**Fig. 6** HS-AFM observations of the non-specific transient binding of Cas9–RNA. **a** Sequential HS-AFM images of Cas9–RNA molecules transiently bound to non-target sites of the DNA. Transient binding of Cas9–RNA appears as a bright spot in the images, as indicated by the white arrows. Most of the binding events were completed within a single-line scanning time (1.9 ms; 150 ms / 80 lines). The scale bar is 50 nm. **b**, **c** Differences between the lowest and highest heights in each image during the HS-AFM observations of DNA only (**b**) and Cas9–RNA–DNA (**c**). In **c**, the spikes indicate the transient binding of Cas9–RNA to the DNA. **d**, **e** Frequency (**d**) and lifetime (**e**) of the transient binding of Cas9–RNA to the DNA. The lifetime was estimated by counting the successive line numbers on which the spike-like spots were continuously seen. Spike-like spots with heights over 3.5 nm were judged as Cas9–RNA molecules. The lifetime was fitted by the 1st order exponential decay, with a time constant of $0.98 \pm 0.02$ ms ($n = 656$)

**Target DNA search by Cas9–RNA**. Previous DNA-curtain assays[11] and single-particle tracking analyses[27] suggested that Cas9–RNA interrogates the target sites via three-dimensional diffusion in vitro and in mammalian cells, respectively. Using HS-AFM, we sought to visualize the target interrogation by the Cas9–RNA complex. However, we failed to observe the movement of Cas9–RNA along the DNA, since the strong interactions between Cas9–RNA and the AP-mica surface suppress the free diffusion of the complexes. In contrast, Cas9–RNA can diffuse more freely on a mica-supported lipid bilayer, thus allowing the HS-AFM observations of the Cas9–RNA movement along the DNA. We adsorbed the 600-bp dsDNA containing a 20-nt target site with the TGG PAM on the mica-supported lipid bilayer, and then added apo-Cas9 or the pre-assembled Cas9–RNA complex (Fig. 5a, b). The HS-AFM movies revealed that multiple apo-Cas9 molecules bind and slide along the DNA (Fig. 5a, Supplementary Movie 6). An analysis of the HS-AFM movies confirmed that apo-Cas9 binds to the DNA in a non-specific manner (Fig. 5c,

Supplementary Fig. 7a), consistent with the DNA-curtain study[11]. A time course analysis of the DNA-bound Cas9 positions confirmed that apo-Cas9 slides along the DNA (Fig. 5d). In contrast, the HS-AFM movies revealed that the Cas9–RNA complexes do not slide along the DNA, and rapidly bind to the target site in a specific manner (Fig. 5b, Supplementary Movie 7). An analysis of the HS-AFM images confirmed the specific binding of Cas9–RNA to the target site (Fig. 5c, d, Supplementary Fig. 7b).

Intriguingly, we observed short-lived bright spots (less than 3 ms) on the DNA (Fig. 6a, Supplementary Movie 8). These spots on the DNA were only observed in the presence of the Cas9–RNA complex, but not in its absence (Fig. 6b–d), suggesting that the observed short-lived spots represent the transient binding of Cas9–RNA to non-target sites. The lifetime of the non-target binding was estimated to be ~1 ms (Fig. 6e). Given that this lifetime is much shorter than the reported value (~3.3 s) from the DNA-curtain study[11], it is possible that the dissociation of the Cas9–RNA complex was facilitated by the contacts with the AFM probe. On the basis of these HS-AFM data, we conclude that Cas9–RNA searches for the target sites by three-dimensional diffusion, rather than one-dimensional sliding, consistent with the DNA-curtain study[11].

## Discussion

HS-AFM enables the direct visualization of the structures and dynamics of intact molecules, in contrast to other single-molecule imaging methods, in which a molecule of interest must be labeled with fluorescent probes. Using HS-AFM, we visualized the real-space and real-time dynamics of CRISPR-Cas9 in action, thereby improving our mechanistic understanding of the RNA-guided DNA cleavage by Cas9. Although our HS-AFM data are essentially consistent with previous fluorescence-based imaging studies[11,17,18], there are some discrepancies between these studies. For example, a recent single-molecule FRET study reported that the majority (90%) of Cas9–RNA stably adopt the D state upon DNA binding, while only 3% of the complexes undergo transitions between the I and D states[18]. In contrast, our HS-AFM data showed that most of the Cas9–RNA complexes fluctuate between the I and D states after DNA binding. This discrepancy is likely due to the differences in the imaging techniques and experimental conditions. On the basis of our HS-AFM data, together with previous structural, biochemical and biophysical data[11–18,28], we propose a model for the Cas9–RNA-mediated DNA cleavage (Fig. 7a–c). Apo-Cas9 adopts a flexible modular architecture, and assembles with the guide RNA to form the stable Cas9–RNA effector complex. The Cas9–RNA complex interrogates the target sites on the DNA via three-dimensional diffusion, and recognizes the complementary target site with the NGG PAM. Cas9–RNA then unwinds the dsDNA target to form the R-loop. The HNH domain undergoes conformational fluctuations upon R-loop formation, and then adopts the catalytically-active docked conformation to cleave the target strand, while the RuvC domain cleaves the non-target strand. Overall, this study provides unprecedented details about the functional dynamics of CRISPR-Cas9, and highlights the potential of HS-AFM to elucidate the action mechanisms of RNA-guided effector nucleases from distinct CRISPR-Cas systems[29].

## Methods

**Sample preparation**. Wild-type *S. pyogenes* Cas9 and GFP-dCas9(D10A/C81L/C574E/H840A) were expressed in *Escherichia coli* Rosetta2 (DE3), and then purified to homogeneity by column chromatography, as described with minor modifications[15]. Briefly, the Cas9 protein was expressed in *E. coli* Rosetta2 (DE3) (Novagen), and was purified by chromatography on Ni-NTA Superflow (QIAGEN), HiTrap SP HP (GE Healthcare), and HiLoad Superdex 200 16/60 (GE Healthcare) columns. The 98-nt guide RNA was transcribed in vitro,

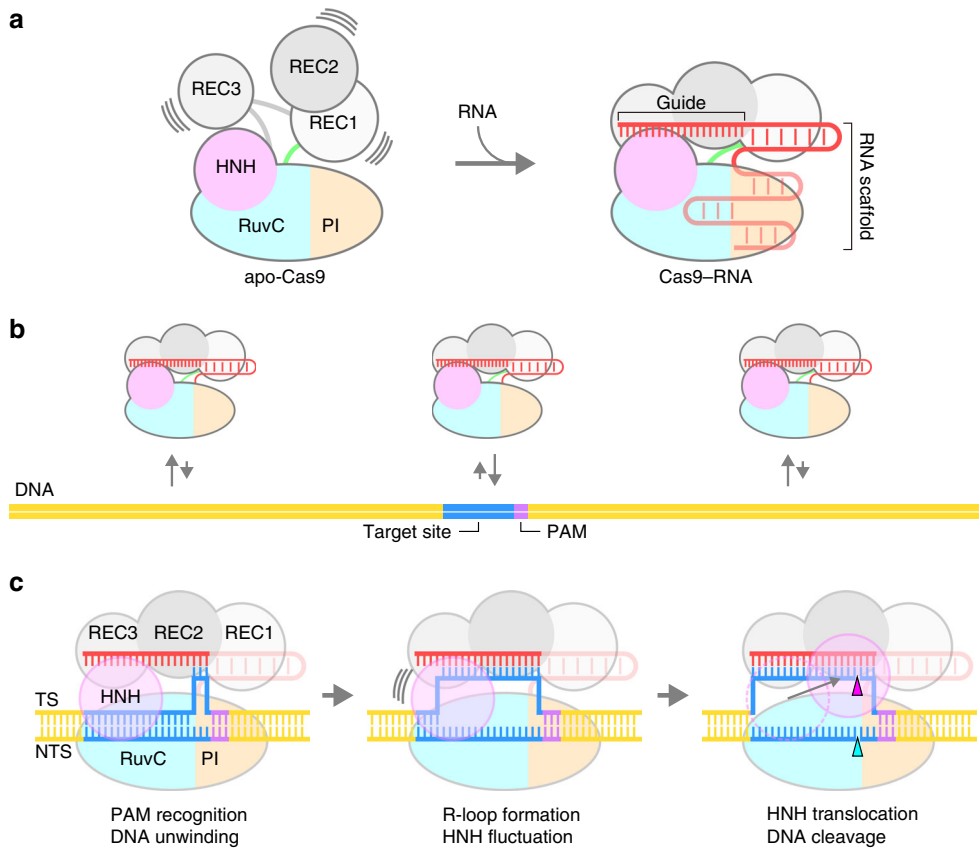

**Fig. 7** Model of CRISPR-Cas9 action. **a** Assembly of the Cas9–RNA binary complex. **b** Target DNA interrogation by Cas9–RNA via three-dimensional diffusion. **c** DNA recognition and cleavage by Cas9–RNA. DNA cleavages by the RuvC and HNH domains are indicated by the cyan and magenta triangles, respectively

and then purified by denaturing urea polyacrylamide gel electrophoresis, as described[15]. The 600-bp target DNA was PCR-amplified using the pUC119 plasmid containing the 20-nt target sequence and the TGG PAM[30] as the template, and then purified using a Wizard DNA Clean-Up System (Promega). The sequences of Cas9, the guide RNA and the target DNA are provided in Supplementary Data 1.

**HS-AFM observations on an AP-mica surface.** The laboratory-built high-speed AFM was used in the tapping mode[31]. The cantilever deflection was detected with an optical beam deflection detector, on which a 0.7 mW, 780 nm infrared laser was mounted. The infrared laser beam was focused onto the back side of the cantilever (Olympus: BL-AC7DS-KU4) through a ×60 objective lens (Nikon: CFI S Plan Fluor ELWD 60×). The reflected laser from the cantilever was detected with a two-segmented PIN photodiode. The spring constant of the cantilever was ~100 pN nm⁻¹. The resonant frequency and the quality factor of the cantilever in liquid were ~800 kHz and ~2, respectively. An amorphous carbon tip was fabricated on the original AFM tip by electron beam deposition (EBD). The length of the additional AFM tip was ~500 nm, and the radius of the apex of the tip was ~4 nm. The free oscillation amplitude of the cantilever was ~1 nm and the set-point amplitude was set to 90% of the free amplitude. For HS-AFM observations of Cas9, a mica surface was treated for 3 min with 0.011% (3-aminopropyl)triethoxysilane (APTES) (Sigma-Aldrich). The complex of Cas9, RNA and DNA was pre-assembled (Cas9:RNA:DNA = 1:1:1 mole ratio) in AFM-imaging buffer. HS-AFM observations of apo-Cas9 and Cas9–RNA were performed in buffer, consisting of 20 mM Tris-HCl, pH 8.0, 100 mM KCl and 0.01 mM EDTA. HS-AFM observations of the Cas9–RNA–DNA and GFP-dCas9–RNA–DNA complexes were performed in buffer, consisting of 20 mM Tris-HCl, pH 8.0, 30 mM KCl and 0.01 mM EDTA. All HS-AFM experiments were performed at room temperature.

**HS-AFM observations on a lipid bilayer.** The DNA was loosely immobilized on a mica-supported lipid bilayer[31]. In brief, 1,2-dipalmitoyl-*sn*-glycero-3-phosphocholine (DPPC; Avanti Polar Lipids) and 1,2-dipalmitoyl-3-trimethylammonium-propane (DPTAP; Avanti Polar Lipids) were suspended in chloroform, and 1,2-dipalmitoyl-*sn*-glycero-3-phosphoethanolamine-*N*-(cap biotinyl) (biotin-cap-DPPE; Avanti Polar Lipids) was suspended in a mixture of chloroform:methanol: water = 65:35:8 (volume ratio). The lipid solutions were mixed at a weight ratio of DPPC:DPTAP:biotin-cap-DPPE = 90:5:5. Small unilamellar vesicles (SUVs) of the

lipid mixture, at 0.2 mg ml⁻¹ in 10 mM MgCl₂, were prepared by sonication for 1 min, using a bus-sonicator (AS ONE). A mica disk (1.0 mm in diameter) glued on the sample stage of the HS-AFM was freshly cleaved, and the mica-supported lipid bilayer was formed on it by depositing 2 μl of the SUVs solution for 3 min, followed by 1 μl of 20 mM Tris-HCl, pH 8.8, for over 10 min. The sample surface was rinsed with drops of Milli-Q water (20 μl × 5) to remove the excess SUVs, and then the solution was replaced with buffer A (20 mM Tris-HCl, pH 8.0, 30 mM KCl, 0.01 mM EDTA). DNA (2 μl, 1 ng μl⁻¹) in buffer A was deposited on the surface for 3 min. After rinsing the surface with buffer B (20 mM Tris-HCl, pH 8.0, 30 mM KCl, 2 mM MgCl₂), the sample stage was immersed in a liquid cell filled with buffer B (~55 μl), and HS-AFM observations were performed. During the HS-AFM observations, a drop (~5 μl) of either apo-Cas9 or Cas9–RNA (molar ratio of Cas9: RNA = 1:2) was added to the liquid cell, at a final concentration of ~80 nM.

**Correlation analysis of HS-AFM images.** 2D correlation coefficients were calculated between the HS-AFM images of the first frame and each of the frames within the Region of Interest (ROI) (i.e., the first frame is the reference)[23]. The sizes of the ROIs for apo-Cas9 and Cas9–RNA that enclosed the whole Cas9 molecule were about 27 × 24 nm². For the Cas9–RNA–DNA complex, the sizes of the ROIs for the REC, HNH and RuvC-PI domains that enclosed the whole region of each domain were about 13 × 10 nm², 7 × 7 nm² and 13 × 10 nm², respectively. The 2D correlation coefficient was calculated frame by frame for each ROI. The 2D correlation coefficient is defined as,

$$r = \frac{\sum_m \sum_n (H_{mn} - \overline{H})(R_{mn} - \overline{R})}{\sqrt{\left(\sum_m \sum_n (H_{mn} - \overline{H})^2\right)\left(\sum_m \sum_n (R_{mn} - \overline{R})^2\right)}}$$

in which $H_{mn}$ and $R_{mn}$ are the heights at the pixel point $(m, n)$ in the ROI to be analyzed and the reference ROI of the reference frame, respectively. $\overline{H}$ and $\overline{R}$ are the mean values of the height matrices $H$ and $R$, respectively.

**Calculation of the center positions of HS-AFM images.** The center positions of the HNH and REC domains were calculated, as described below. First, the ROIs that enclosed the whole HNH and REC domains were about 7 × 7 nm², and then the center positions of the ROIs were calculated by the X, Y, and Z data of the HS-AFM images. The X and Y data correspond to the lateral coordinates, while the

Z data correspond to the height. After the conformational change in the HNH domain, the identical ROIs were used for the calculation of the center positions.

**Data availability**. The data sets generated during the current study are available from the corresponding author upon request.

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

## Acknowledgements

This work was supported by the Kao Foundation for Arts and Science (M.S.), the Brain Science Foundation (M.S.), JST/PRESTO (H.N. (JPMJPR13L8) and N.K. (JPMJPR13L4)), JST/CREST (T.A. (JPMJCR13M1)), the Basic Science and Platform Technology Program for Innovative Biological Medicine from the Japan Agency for Medical Research and Development, AMED (O.N.), and JSPS KAKENHI Grant Numbers JP16K18523 (M.S.), 26291010 and 15H01463 (H.N.), 15H04360 (N.K.), 24227005 and 26119003 (T.A.), and 16H00830, 16H00758 and 15H03540 (T.U.).

## Author contributions

M.S. and H.N. conceived and designed the experiments; M.S. performed HS-AFM experiments on the AP-mica surface; N.K. performed HS-AFM experiments on the lipid bilayer; M.S. analyzed the data; H.N. and S.H. prepared Cas9, RNA and DNA; T.A. developed HS-AFM techniques; T.U. developed the software for the HS-AFM instrument and data analysis; M.S. and H.N. wrote the paper; and T.A., O.N. and T.U. supervised the project. All authors discussed the results and commented on the manuscript.

## Additional information

**Competing interests:** The authors declare no competing financial interests.

