## [Peer Review File · Nature Communications]

Reviewers' Comments:

Reviewer #1 (Remarks to the Author):

The paper « Real-space and real-time dynamics of CRISPR-Cas9 visualized by high-speed atomic force microscopy » by Mikihiro Shibata, Hiroshi Nishimasu, Noriyuki Kodera, Seiichi Hirano, Toshio Ando, Takayuki Uchihashi, and Osamu Nureki, describes a high-speed atomic force microscopy study of the structure and dynamics of Cas9 alone and in interaction with RNA and DNA.

The topic is of high importance, the technique cutting edge and the data of highest quality. As such the paper deserves publication.

My main concern is the writing of the paper. Some parts of the paper contain so many jargon expressions related to the Cas9 system that it becomes incomprehensible. Of course the CRISPR-Cas9 system attracts these days high attention, but to make the paper understandable for readers outside of this field it needs careful rewriting/simplification/explanation of terms.

Abstract: « RNA-guided » « dual-guide » « chimeric single guide RNA ». Maybe, after reading the paper, a scientist that does not work with CRISPR-Cas does understand these expressions. However, probably not before. The authors should make an effort to reformulate their abstract in a way that it becomes attractive for non-specialists too.

Abstract: « HS-AFM movies... ..three-dimensional diffusion ». After reading the paper to the end I seem to understand that this is an interpretation based on the fact that the authors see dissociation events from DNA and reason that in the native system the search is 3D. The way it is written is however confusing as the data does not show 3D diffusion, which is by the way probably impossible to see by a surface technique like AFM.

Introduction: « dual guide RNAs (CRISPR RNA and trans-activating CRISPR RNA) » « protospacer adjacent motif (PAM) » « crRNA » « R-loop formation » « HNH and RuvC nuclease domains » are expressions only for specialists. Abbreviations like crRNA are not defined. All these expressions appear before the first mention of an explanation figure panel.

Introduction: HS-AFM-refs. Recent HS-AFM references from other groups, i.e. Chiaruttini, Cell 2015, Miyagi, Nature Nano 2016, could be cited.

Results: line 85: « apo-Cas9 adopts flexible modular conformations, unlike the crystal structure of apo-Cas9 ». Of course. The crystal structure by definition does not reveal flexibility. This does however not exclude that the protein is not flexible.

Results: « A structural comparison between apo-Cas9 and Cas9–RNA indicated »... The authors show clearly that the apo-Cas9 is very different from the Cas9-RNA. Interestingly there is not a single frame in which the apo-Cas9 resembles the Cas9-RNA, which would mean that the Cas9-RNA interaction leads to a conformation that the apo form never adopts! Is it induced fit?

Results: « adopt flexible conformations... although the crystal structure ». Again, the crystal structure can only depict a single conformation. The flexibility is not the key issue here. I would rather suggest that an interesting aspect is that the apo protein in HS-AFM never seems to adopt the structure that it has in the crystal.

Results: PAM-dependent... paragraph: « TGG PAM » « TTT PAM » « NGG PAM ». Please explain.

Same paragraph and elsewhere: The authors often gather several panels into one figure citation, e.g. « The HS-AFM movies... ..(Fig 2a-2d) ». Only one panel of the 4 displays movie frames. The other panels contain analysis. The paper would read better if each panel would be briefly stated and explained.

Results: Conformational dynamics... paragraph: The lower state of the HNH domain is a detailed assignment with an amplitude change of 8Å on a globular not necessarily orientationally fixed molecule. Was the height measurement the criterion for the assignment of the state or was the DNA cleavage afterwards the criterion for deciding when the HNH started to adopt the lower state. Also the authors state that the activated state could have very different lifetimes until the DNA dissociates, how can we then tell from which moment onwards the HNH-lowered conformation is adopted.

Results: Target DNA... paragraph and elsewhere: « Previous imaging studies suggested ». What imaging studies? By what technique? HS-AFM?

Results: last sentence. The three-dimensional search mode is an interpretation for how the molecule works in vivo, based on observed dissociation events in HS-AFM. Clarify.

Altogether, a nice paper with excellent data, but the writing makes it sometimes difficult to appreciate the impact for non-specialists.

Reviewer #2 (Remarks to the Author):

In this manuscript, Shibata et al. visualize Cas9 target searching and conformational dynamics using high-speed atomic-force microscopy (HS-AFM). HS-AFM is a powerful tool for directly visualizing macromolecules in real-time and real-space, and the data presented in both figures and especially movies are compelling and beautifully illustrate the previously characterized conformational flexibility of Cas9. This conformational flexibility has important implications for the fidelity of Cas9 cleavage, and has been the subject of several recent studies using ensemble and single-molecule FRET (Sternberg et al. *Nature*, 2015, Dagdas et al. *bioRxiv*, 2017, Osuka et al. *bioRxiv*, 2017, Chen et al. *bioRxiv*, 2017). The HS-AFM data presented in this manuscript are consistent with these previous studies. In addition, the authors visualize Cas9 target binding in real time, including non-specific binding at off-target sites. These experiments follow previous studies using DNA curtains (Sternberg et al. *Nature*, 2014), providing similar insight into the Cas9 target searching mechanism.

Overall, the results and conclusions of these investigations are consistent with previous biophysical studies, although it is unclear whether the study provides new insights beyond what is presented in those previous studies. Nevertheless, the application of HS-AFM to directly visualize Cas9 target searching and binding is exciting and worthy of publication in *Nature Communications*.

Minor comments:

1. It would be interesting if possible to include information on the percentage of Cas9 molecules that underwent fluctuations of the HNH domain upon dsDNA binding. In an smFRET study of HNH movement, Dagdas et al. observed only 3% of molecules undergo transitions between the intermediate and docked state, while 90% of molecules stably adopted the docked state. A comparison of results between these studies would be a useful addition to the Discussion section. In addition, the authors may consider comparing their off-target dwell time analysis with that of the previous DNA curtains study (Sternberg et al. *Nature*, 2014).
2. On page 5, line 113 the citation for Figure 2e should likely be Figure 2f.

Reviewer #3 (Remarks to the Author):

The structure-function analysis of CRISPR-Cas9 has attracted enormous general interest because 1) the knowledge can be directly transferred to various genome editing applications and 2) this has become a model system for the study of protein-RNA complex assembly, conformation

dynamics, RNA-guided nucleic acid recognition, nuclease activation, etc. This manuscript by Shibata et. al used the high-speed atomic force microscopy (HS-AFM) technique to reveal the conformational dynamics involved in gRNA binding, target searching, and nuclease activation. Specifically, they could directly visualize the movement of the catalytic HNH domain and its correlation with the DNA cleavage event. The observation that Cas9 releases DNA after cleavage challenges the main conclusion in the 2015 Nature study by Sternberg et al, and should inspire follow-up studies to probe the Cas9 mechanism further. I find the results well-controlled and the interpretations are convincing (a few minor reservations are listed below). The messages from this study are novel, and would fit the general readership of Nature Communications.

1. The stronghold of the AFM method is that it is a directly observation tool. The researchers do not rely on FRET or force changes to infer the mechanism. The weakness is that AFM scanning exerts strong forces on the macromolecule, hence it could induce artificial movements in some cases. Although based on the listed controls, the authors certainly had these understanding in mind when doing the experiments, this reviewer suggests that a few sentences should be included in the discussion to compare and contrast the AFM and other single molecule methods in the study of Cas9 system.
2. An important conclusion of the manuscript is that HNH moves to a new location upon target-binding. This conclusion is mainly based on the observation of a protrusion in the AFM image. Given that the resolution of the AFM is rather limited, how unique is the domain assignment? To fully convince the readers, shouldn't the authors do one more control, to delete HNH from Cas9 and show that the protrusion now disappears?
3. Fig. 3d, the authors show that the cleavage product dissociates mostly from the non-PAM side, but in 25% of the cases, it could also dissociate from the PAM-side. This would require the complete unwinding of the gRNA-target DNA duplex. The authors should speculate how this could take place. Moreover, this observation contradicts the 2015 Nature study by Sternberg et al, which should be adequately discussed as well.
4. Did the authors observe any local DNA bending when Cas9-gRNA complex is binding to the canonical or non-canonical target site (Figs. 3a-b versus 6a and c)?

Responses to the reviewers' comments

Reviewer #1 (Remarks to the Author):

The paper « Real-space and real-time dynamics of CRISPR-Cas9 visualized by high-speed atomic force microscopy » by Mikihiro Shibata, Hiroshi Nishimasu, Noriyuki Kodera, Seiichi Hirano, Toshio Ando, Takayuki Uchihashi, and Osamu Nureki, describes a high-speed atomic force microscopy study of the structure and dynamics of Cas9 alone and in interaction with RNA and DNA. The topic is of high importance, the technique cutting edge and the data of highest quality. As such the paper deserves publication. My main concern is the writing of the paper. Some parts of the paper contain so many jargon expressions related to the Cas9 system that it becomes incomprehensible. Of course the CRISPR-Cas9 system attracts these days high attention, but to make the paper understandable for readers outside of this field it needs careful rewriting/simplification/explanation of terms.

We thank the reviewer for the positive comments and helpful advice. According to the reviewer's suggestions, we carefully revised the text and the figures so that the readers outside of the CRISPR field can understand it more readily.

Abstract: « RNA-guided » « dual-guide » « chimeric single guide RNA ». Maybe, after reading the paper, a scientist that does not work with CRISPR-Cas does understand these expressions. However, probably not before. The authors should make an effort to reformulate their abstract in a way that it becomes attractive for non-specialists too.

We have revised the abstract, according to the reviewer's suggestions.

Abstract: « HS-AFM movies... ...three-dimensional diffusion ». After reading the paper to the end I seem to understand that this is an interpretation based on the fact that the authors see dissociation events from DNA and reason that in the native system the search is 3D. The way it is written is however confusing as the data does not show 3D diffusion, which is by the way probably impossible to see by a surface technique like AFM.

We have revised the abstract, according to the reviewer's suggestions.

Introduction: « dual guide RNAs (CRISPR RNA and trans-activating CRISPR RNA) » « protospacer adjacent motif (PAM) » « crRNA » « R-loop formation » « HNH and RuvC nuclease domains » are expressions only for specialists. Abbreviations like crRNA are not defined. All these expressions appear before the first mention of an explanation figure panel.

We have revised the introduction, according to the reviewer's suggestions. In addition, we have included a new figure showing a schematic description of the DNA cleavage by Cas9–RNA, in Supplementary Fig. 1a.

Introduction: HS-AFM-refs. Recent HS-AFM references from other groups, i.e. Chiaruttini, Cell 2015, Miyagi, Nature Nano 2016, could be cited.

We have cited these references in the revised manuscript.

Results: line 85: « apo-Cas9 adopts flexible modular conformations, unlike the crystal structure of apo-Cas9 ». Of course. The crystal structure by definition does not reveal flexibility. This does however not exclude that the protein is not flexible.

We agree with the reviewer's comments, and we think that the closed conformation of apo-Cas9 observed in the crystals structure is stabilized through crystal packing interactions, as mentioned in the manuscript.

Results: « A structural comparison between apo-Cas9 and Cas9–RNA indicated »... The authors show clearly that the apo-Cas9 is very different from the Cas9-RNA. Interestingly there is not a single frame in which the apo-Cas9 resembles the Cas9-RNA, which would mean that the Cas9-RNA interaction leads to a conformation that the apo form never adopts! Is it induced fit?

The guide-RNA binding stabilizes the Cas9 conformation, thereby facilitating the effector complex formation, as described in the manuscript.

Results: « adopt flexible conformations... although the crystal structure ». Again, the crystal structure can only depict a single conformation. The flexibility is not the key issue here. I would

rather suggest that an interesting aspect is that the apo protein in HS-AFM never seems to adopt the structure that it has in the crystal.

In the HS-AFM images, we detected some apo-Cas9 molecules adopting a closed conformation similar to that observed in the crystal. Nonetheless, it is unclear whether they adopt the same closed conformation, due to difficulty in the precise assignment of each domain in the HS-AFM images.

Results: PAM-dependent... paragraph: « TGG PAM » « TTT PAM » « NGG PAM ». Please explain.

We have included the explanations for the PAM in the text and the figures.

Same paragraph and elsewhere: The authors often gather several panels into one figure citation, e.g. « The HS-AFM movies... ... (Fig 2a-2d) ». Only one panel of the 4 displays movie frames. The other panels contain analysis. The paper would read better if each panel would be briefly stated and explained.

In the revised manuscript, we have separately explained each figure panel as much as possible.

Results: Conformational dynamics... paragraph: The lower state of the HNH domain is a detailed assignment with an amplitude change of 8Å on a globular not necessarily orientationally fixed molecule. Was the height measurement the criterion for the assignment of the state or was the DNA cleavage afterwards the criterion for deciding when the HNH started to adopt the lower state. Also the authors state that the activated state could have very different lifetimes until the DNA dissociates, how can we then tell from which moment onwards the HNH-lowered conformation is adopted.

We used the height measurement as the criterion of the assignment of the functional state.

Results: Target DNA... paragraph and elsewhere: « Previous imaging studies suggested ». What imaging studies? By what technique? HS-AFM?

We have specifically described the imaging techniques, such as the DNA-curtain assay, bulk FRET, and smFRET.

Results: last sentence. The three-dimensional search mode is an interpretation for how the molecule works in vivo, based on observed dissociation events in HS-AFM. Clarify.

We have modified the text, according to the reviewer's comments.

Altogether, a nice paper with excellent data, but the writing makes it sometimes difficult to appreciate the impact for non-specialists.

We thank the reviewer for the positive comments and helpful suggestions.

Reviewer #2 (Remarks to the Author):

In this manuscript, Shibata et al. visualize Cas9 target searching and conformational dynamics using high-speed atomic-force microscopy (HS-AFM). HS-AFM is a powerful tool for directly visualizing macromolecules in real-time and real-space, and the data presented in both figures and especially movies are compelling and beautifully illustrate the previously characterized conformational flexibility of Cas9. This conformational flexibility has important implications for the fidelity of Cas9 cleavage, and has been the subject of several recent studies using ensemble and single-molecule FRET (Sternberg et al. Nature, 2015, Dagdas et al. bioRxiv, 2017, Osuka et al. bioRxiv, 2017, Chen et al. bioRxiv, 2017). The HS-AFM data presented in this manuscript are consistent with these previous studies. In addition, the authors visualize Cas9 target binding in real time, including non-specific binding at off-target sites. These experiments follow previous studies using DNA curtains (Sternberg et al. Nature, 2014), providing similar insight into the Cas9 target searching mechanism.

Overall, the results and conclusions of these investigations are consistent with previous biophysical studies, although it is unclear whether the study provides new insights beyond what is presented in those previous studies. Nevertheless, the application of HS-AFM to directly visualize Cas9 target searching and binding is exciting and worthy of publication in Nature Communications.

We thank the reviewer for the positive comments.

Minor comments:

1. It would be interesting if possible to include information on the percentage of Cas9 molecules that underwent fluctuations of the HNH domain upon dsDNA binding. In an smFRET study of HNH movement, Dagdas *et al.* observed only 3% of molecules undergo transitions between the intermediate and docked state, while 90% of molecules stably adopted the docked state. A comparison of results between these studies would be a useful addition to the Discussion section. In addition, the authors may consider comparing their off-target dwell time analysis with that of the previous DNA curtains study (Sternberg *et al.* *Nature*, 2014).

In the revised manuscript, we have discussed the different ratios between the I and D states between our HS-AFM study and a recent smFRET study (Dagdas *et al.* *Sci. Adv.* 2017). We have also mentioned the differences in the off-target dwell times between our study and the previous DNA-curtain study (Sternberg *et al.* *Nature* 2014).

2. On page 5, line 113 the citation for Figure 2e should likely be Figure 2f.

We have revised the citation for the figure.

Reviewer #3 (Remarks to the Author):

*The structure-function analysis of CRISPR-Cas9 has attracted enormous general interest because 1) the knowledge can be directly transferred to various genome editing applications and 2) this has become a model system for the study of protein-RNA complex assembly, conformation dynamics, RNA-guided nucleic acid recognition, nuclease activation, etc. This manuscript by Shibata *et al.* used the high-speed atomic force microscopy (HS-AFM) technique to reveal the conformational dynamics involved in gRNA binding, target searching, and nuclease activation. Specifically, they could directly visualize the movement of the catalytic HNH domain and its correlation with the DNA cleavage event. The observation that Cas9 releases DNA after cleavage challenges the main conclusion in the 2015 *Nature* study by Sternberg *et al.*, and should inspire follow-up studies to probe the Cas9 mechanism further. I find the results well-controlled and the interpretations are convincing (a few minor reservations are listed below). The messages from this study are novel, and would fit the general readership of *Nature Communications*.*

We thank the reviewer for the enthusiastic comments.

1. The stronghold of the AFM method is that it is a directly observation tool. The researchers do not rely on FRET or force changes to infer the mechanism. The weakness is that AFM scanning exerts strong forces on the macromolecule, hence it could induce artificial movements in some cases. Although based on the listed controls, the authors certainly had these understanding in mind when doing the experiments, this reviewer suggests that a few sentences should be included in the discussion to compare and contrast the AFM and other single molecule methods in the study of Cas9 system.

In the Discussion, we have included the statement about the differences between HS-AFM and the other imaging methods.

2. An important conclusion of the manuscript is that HNH moves to a new location upon target-binding. This conclusion is mainly based on the observation of a protrusion in the AFM image. Given that the resolution of the AFM is rather limited, how unique is the domain assignment? To fully convince the readers, shouldn't the authors do one more control, to delete HNH from Cas9 and show that the protrusion now disappears?

Thank you for the helpful suggestions. We performed the domain assignment based on a comparison between the crystal structures and the HS-AFM images of Cas9–RNA–DNA and GFP–dCas9–RNA–DNA. We believe that the domain assignment is sufficiently accurate, although the assignment will be reinforced by HS-AFM data of Cas9 lacking the HNH domain.

3. Fig. 3d, the authors show that the cleavage product dissociates mostly from the non-PAM side, but in 25% of the cases, it could also dissociate from the PAM-side. This would require the complete unwinding of the gRNA-target DNA duplex. The authors should speculate how this could take place. Moreover, this observation contradicts the 2015 Nature study by Sternberg et al, which should be adequately discussed as well.

Although the release of the non-PAM side would require the unwinding of the RNA–DNA heteroduplex, it is unclear how the non-PAM side is released from the complex, due to the limited resolution of the HS-AFM imaging. As the reviewer pointed out, the release of the non-PAM side was not observed in a DNA-curtain assay (Sternberg *et al. Nature* 2014), and this discrepancy may be derived from the effects of the contacts with the AFM probe. We have

included these discussions in the revised manuscript.

4. Did the authors observe any local DNA bending when Cas9-gRNA complex is binding to the canonical or non-canonical target site (Figs. 3a-b versus 6a and c)?

In the HS-AFM movies, we observed local bending in the DNA upon the binding of Cas9–RNA to the target site, consistent with the crystal structures. In contrast, it is unclear whether the binding of Cas9–RNA to non-target sites induces local bending, due to the limited resolution of the HS-AFM observations on a lipid bilayer. In the revised manuscript, we have included a statement about the local bending of the DNA induced by Cas9–RNA binding.